# Identification of Epitopes on Rhinovirus 89 Capsid Proteins Capable of Inducing Neutralizing Antibodies

**DOI:** 10.3390/ijms23095113

**Published:** 2022-05-04

**Authors:** Katarzyna Niespodziana, Clarissa R. Cabauatan, Petra Pazderova, Phyllis C. Vacal, Judith Wortmann, Walter Keller, Peter Errhalt, Rudolf Valenta

**Affiliations:** 1Division of Immunopathology, Department of Pathophysiology and Allergy Research, Center for Patho-Physiology, Infectiology and Immunology, Medical University of Vienna, 1090 Vienna, Austria; katarzyna.niespodziana@meduniwien.ac.at (K.N.); clarissa.cabauatan@gmail.com (C.R.C.); pazderova.petra@gmail.com (P.P.); 2Karl Landsteiner University of Health Sciences, 3500 Krems, Austria; 3Center for Natural Sciences, School of Health and Natural Sciences, Saint Mary’s University, Bayombong 3700, Nueva Vizcaya, Philippines; phyllis.vacal@yahoo.fr; 4Institute of Molecular Biosciences, BioTechMed Graz, University of Graz, 8010 Graz, Austria; judith.wortmann@gmx.de (J.W.); walter.keller@uni-graz.at (W.K.); 5Department of Pneumology, University Hospital Krems and Karl Landsteiner University of Health Sciences, 3500 Krems, Austria; peter.errhalt@krems.lknoe.at; 6NRC Institute of Immunology MBA of Russia, Moscow 115478, Russia; 7Laboratory for Immunopathology, Department of Clinical Immunology and Allergy, Sechenov First Moscow State Medical University, Moscow 119435, Russia

**Keywords:** rhinovirus, neutralizing epitopes, epitope mapping, asthma, vaccine

## Abstract

Rhinoviruses (RVs) are major causes of the common cold, but they can also trigger exacerbations of asthma. More than 160 different RV strains exist and can be classified into three genetic species (RV-A, RV-B and RV-C) which bind to different receptors on human cells including intracellular adhesion molecule 1 (ICAM-1), the low-density lipoprotein receptor (LDLR) or the cadherin-related family member 3 (CDHR3). Epitopes located in the RV capsid have mainly been determined for RV2, a minor-group RV-A strain binding to LDLR, and for RV14, a major-group RV-B strain binding to ICAM-1. In order to study epitopes involved in the neutralization of RV89, an ICAM-1-binding RV-A strain which is highly different from RV2 and RV14 in terms of receptor specificity and sequence, respectively, we analyzed the specificity and epitopes of a highly neutralizing antiserum using recombinantly produced RV89 capsid proteins (VP1, VP2, VP3 and VP4), recombinant fragments and synthetic overlapping peptides thereof. We found that the antiserum which neutralized in vitro RV89 infection up to a dilution of 1:24,000 reacted with the capsid proteins VP1 and VP2 but not with VP3 and VP4. The neutralizing antibodies recognized recombinant fragments comprising approximately 100 amino acids of the N- and C-terminus of VP1 and the middle part of VP2, in particular, three peptides which, according to molecular modeling based on the three-dimensional structure of RV16, were surface-exposed on the viral capsid. Two recombinant fusion proteins containing the identified peptides fused to hepatitis B (HBV)-derived preS as a carrier protein induced upon immunization of rabbits antibodies capable of neutralizing in vitro RV89 infections. Interestingly, the virus-neutralizing epitopes determined for RV89 corresponded to those determined for minor-group RV2 binding to LDL and major-group RV14 belonging to the RV-B species, which are highly different from RV89. Our results indicate that highly different RV strains, even when reacting with different receptors, seem to engage similar parts of their capsid in the infection process. These results may be important for the design of active and passive immunization strategies for RV.

## 1. Introduction

Rhinoviruses (RVs) are major causative agents of the common cold. The symptoms of common cold are usually self-limiting and are usually not severe, simply requiring an ample amount of fluid intake and bedrest. However, common cold leads to a substantial loss of productivity in working-age groups and school attendance in children [1]. Importantly, RV infections are associated with exacerbations of severe respiratory diseases such as asthma among children and adults [2,3,4,5] and chronic obstructive pulmonary disease in the elderly [6,7]. RV-induced exacerbations of asthma in children may be especially severe in those with an atopic background [8]. Furthermore, the detection of pathogenic bacteria during the RV infection has also been shown to be associated with the increased severity of respiratory illnesses, including exacerbations of asthma [9].

RVs belong to the genus Enterovirus of the Picornaviridae family [10]. Their single-stranded positive-sense RNA genome is surrounded by an icosahedral protein capsid comprising 60 copies of four structural proteins, the surface-exposed VP1, VP2, VP3 and the buried VP4 [11]. Since the discovery of RVs in the 1950s, a large number of distinct serotypes of RV have been characterized [12] and classified, initially based on antibody neutralization properties of antisera and then by sequencing [10,13]. In particular, sequence analysis of genomic RV RNA has revealed the existence of more than 160 RV strains that have been classified into three genetic species (RV-A, RV-B and RV-C) [10]. Interestingly, RVs have been shown to bind to different receptors expressed on human airway epithelial cells, but they also enter other cell types such as cells of the immune system [14,15]. The majority of RVs from RV-A and RV-B species use the intracellular adhesion molecule 1 (ICAM-1) as a receptor [16], while a subset of RV-A species belonging to the minor group binds to the low-density lipoprotein receptor (LDLR) [17]. The more recently identified RV-C species has been suggested to use cadherin-related family member 3 (CDHR3) as a receptor for cell entry [18]. This extensive variation among RV strains and the use of different receptors for infection contribute to the difficulty in developing cross-protective RV vaccines and inducing antibodies blocking the binding of various RVs to their receptors [19,20]. In this context, it should be mentioned that the analysis of RV-specific antibody responses of children experiencing RV-induced wheezing exacerbations suggest that RVs belonging to species A, B and C, as well as major- and minor-group RVs, may induce asthma exacerbations [21]. 

Accordingly, there have been several efforts to develop RV vaccines capable of inducing neutralizing antibody responses. In the late 1960s and early 1970s, formalin inactivated whole-virion RVs were administered to humans in several studies [22,23,24,25]. Alternative approaches used conserved RV proteins or peptides as vaccine antigens to immunize animals and to induce cross-reactive antibody responses neutralizing different RV species [26,27]. One study used synthetic peptides corresponding to immunogenic sites of VP1, VP2 and VP3 of the minor-group RV2 strain [28]. Using this approach, a peptide, designed as NImII derived from the VP2 capsid protein, was found to elicit antibodies that neutralized virus infectivity [28] (Table 1). Interestingly, the neutralizing capacity of this region has also been demonstrated in other studies for the major-group RV14 strain [29,30]. Table 1 provides a brief summary of amino acids and peptides which have been mapped, mainly for minor-group and major-group RV strains, in the past. In this context, it is interesting that earlier epitope mapping experiments identified epitopes on minor-group and major-group RV strains which are located in similar regions on VP1 and VP2 capsid proteins, although these RV strains differ considerably in their sequence and utilize different receptors. 

In this study, we mapped epitopes recognized by a highly neutralizing antiserum in the capsid of an RV89 strain belonging to the major-group A virus species that has been shown to be frequently recognized by children suffering from asthma exacerbations triggered by RV infections [21,38]. Furthermore, it could also be shown that antibodies induced with the recombinant VP1 protein of RV89 exhibited cross-neutralizing properties against several RV strains [37]. Identified peptide epitopes were then expressed as fusion proteins together with hepatitis virus B (HBV)-derived preS to raise antibodies capable of neutralizing the infection of cultured human cells by RV89 in vitro. Our experiments identify virus-neutralizing epitopes located on VP1 and VP2 of RV89, which overlap with neutralizing epitopes determined for RV14 group-B major RV, which highly differs in sequence from RV89 and RV2, a minor-group RV strain using a receptor (i.e., LDLR) which is different from ICAM-1 used by RV89 and RV14. Our data thus indicate the presence of conserved receptor-binding regions in VP1 and VP2 of highly different RV strains. 

## 2. Results

### 2.1. Identification of an Antiserum with High Titers of RV89-Neutralizing Antibodies

We have previously found that a guinea pig antiserum raised against RV89 is highly potent in neutralizing RV infections in vitro [37]. We therefore investigated to what extent the guinea pig anti-RV89 antiserum can protect HeLa cells from an infection with rhinovirus in vitro. Results from a representative experiment performed with the RV89 strain are shown in Figure 1. When cells were infected with a 50-fold tissue culture infective dose (TCID_50_) of RV89, a complete cytopathic effect was observed (Figure 1, row: RV89). The addition of serially diluted anti-RV89 guinea pig antiserum prevented RV-induced cell death up to a dilution of 1:24,000 (Figure 1, GPαRV89 + RV89 rows: 1 to 4). Cells incubated in medium without the virus were fully alive (Figure 1, row medium). Addition of normal guinea pig serum to the cells showed no protection against infection (Figure 1, normal GP + RV89 rows 1 to 4). Accordingly, the GPαRV89 antiserum was identified as a highly neutralizing antiserum. 

### 2.2. Neutralizing Antibodies React with Three Protein Fragments of VP1 and VP2 Capsid Proteins but Neither with VP3 Nor with VP4

Having identified GPαRV89 as a highly neutralizing antiserum, we then studied the capsid proteins and epitopes of RV89 which are recognized by this antiserum, to identify epitopes which may be involved in the neutralization of the virus. For this purpose, we produced a repertoire of recombinant capsid proteins, protein fragments and synthetic peptides of RV89. The scheme in Figure 2 provides an overview of the rhinovirus genome encoding four capsid proteins, designated VP1–VP4, and seven replication proteins (2A: protease; 2B: membrane-associated protein; 2C: NTPase; 3A: exact function not yet known; 3B: protein primer for viral RNA synthesis; 3C: protease; 3D: polymerase). The coding region is surrounded by the 5′ UTR containing the IRES and the sequence for the VPg protein, as well as the 3′ UTR containing a poly(A) tail (Figure 2). The magnifications in Figure 2 show VP1 (top) and VP2 (bottom) with the recombinant fragments (i.e., PI, PII, PIII) thereof, each comprising approximately 100 amino acids produced as fusion proteins with maltose binding protein (MBP), as well as synthetic peptides spanning VP1 and VP2 (i.e., VP1-p1-9; VP2-p1-p17). Appendix A provides a summary of the recombinant VP1 and VP2 fragments and the synthetic peptides used for the epitope mapping experiments. 

Figure 3A shows the Coomassie-stained SDS-PAGE containing the purified recombinant proteins, which exhibited the calculated molecular masses. We then blotted the recombinant RV proteins onto a nitrocellulose membrane and analyzed their reactivity with GPαRV89. We found that GPαRV89 reacted with blotted VP1 and VP2 but not with VP3 and VP4 (Figure 3B). No reactivity to the unrelated control protein, PreS-P1-P5, was observed (Figure 3B). Therefore, in the next set of experiments, we analyzed the reactivity of GPαRV89 towards recombinant VP1 and VP2 fragments, and for control purposes, MBP (Figure 3B, middle and right panels). GPαRV89 reacted specifically with two 89VP1 fragments, PI and PIII, and one 89VP2 fragment, PII, respectively. No reactivity to MBP (negative control) was observed (Figure 3B, middle and right panels). Results obtained by the immunoblotting experiments were confirmed by ELISA experiments, in which recombinant capsid proteins and fragments thereof were immobilized onto an ELISA plate and incubated with different dilutions of GPαRV89 antibodies (Figure 3C). GPαRV89 antibodies reacted with 89VP1, 89VP2, 89VP1-PI, 89VP1-PIII and 89VP2-PII. In the ELISA experiments, a very weak reactivity at the lowest dilution (i.e., 1:500) of GPαRV89 with 89VP3 was noted (Figure 3C). 

### 2.3. GPαRV89 Virus-Neutralizing Antibodies Recognize Peptide Epitopes in VP1 and VP2 

In the next set of experiments, we tested GPαRV89 antibodies for reactivity with synthetic peptides spanning the two capsid proteins VP1 and VP2 (Figure 2, Appendix A). GPαRV89 antibodies recognized isolated RV89 as well as 89VP1 and 89VP2 in a dilution-dependent manner (Figure 4). Furthermore, GPαRV89 antibodies also reacted with 89VP1-p1, 89VP1-p3 and 89VP1-p9 from VP1 and with 89VP2-p11 from VP2 in a dilution-dependent manner (Figure 4). Although 89VP1-p5 reportedly contains amino acids which were suggested to be involved in virus neutralization (Appendix A) [11,33], GPαRV89 did not bind to this peptide. The somewhat lower reactivity with 89VP2-p11 may be due to the fact that peptide 11 was shorter than the VP1-derived peptides (Appendix A). No reactivity to any of the other VP1- and VP2-derived peptides was found (data not shown). Likewise, no reactivity was observed to either 89VP3, 89VP4 or to BSA or MBP in this experiment (Figure 3B, data not shown).

In order to visualize the position of the recognized VP1 and VP2 proteins and peptide epitopes on the three-dimensional rhinovirus capsid, we used the only available structure of RV16, which is also a member of group A RV strains closely related to RV89 (sequence identities of 66% and 78% for VP1 and VP2, respectively). Figure 5 shows a surface representation of the RV16 capsid when seen from either the outside (left image) or the inside (right image) of the capsid. As demonstrated by molecular modelling, VP1, VP2 and VP3 were located at the outer part of the surface, whereas VP4 was located in the inner part of the capsid (Figure 5A). Furthermore, 89VP1-p3, 89VP1-p5, 89VP1-p9 and 89VP2-p11 were surface-exposed, whereas the N-terminal 89VP1-p1 was located inside of the virus capsid, as previously reported [39]. While 89VP1-p3, 89VP1-p9 and 89VP2-p11 were identified based on the epitope mapping, 89VP1-p5 was included in the construction of recombinant antigens for the induction of neutralizing antibodies as it has been shown to include amino acids involved in receptor binding [11].

### 2.4. Recombinant PreS-Fusion Proteins Containing VP1 and VP2 Epitopes Recognized by GPαRV89 Induce Protective Antibody Responses upon Immunization

According to the epitope mapping performed with GPαRV89, several VP1- and VP2-derived peptides (i.e., 89VP1-p1, 89VP1-p3, 89VP1-p9 and 89VP2-p11) were identified which may be involved in virus neutralization. According to the structural modelling, peptide 89VP1-p1 appeared to be located inside the capsid and was not accessible on the surface of the virus. Moreover, we previously found that rabbit anti-89VP1-p1 antibodies did not cause virus neutralization (data not shown). Therefore, 89VP1-p1 was not considered to be involved in virus neutralization. However, 89VP1-p5 reportedly contains amino acids involved in virus neutralization [11,33], and appears in close vicinity to 89VP1-p3 in the model of the virus surface. Accordingly, we hypothesized that 89VP1-p3, 89VP1-p5, 89VP1-p9 and 89VP2-p11 could be epitopes involved in virus neutralization. In order to test the ability of these peptides to induce virus-neutralizing antibody responses upon immunization, we produced two recombinant fusion proteins (i.e., 89RX1 and 89RX2) consisting of these peptides and the HBV-derived PreS protein as a carrier to render the peptides immunogenic. In 89RX1, two copies of 89VP1-p3 were attached to the N- terminus and two copies of 89VP2-p11 to the C-terminus of PreS, whereas in 89RX2, two copies of 89VP2-p11 were fused to the N-terminus and one copy of 89VP1-p5 and 89VP1-p9 to the C-terminus of PreS (Figure 6A). Furthermore, 89RX1 and 89RX2 were expressed in *E. coli*, purified and used to immunize rabbits.

Anti-89RX1 and anti-89RX2 antibodies showed a comparable reactivity to natural RV89 (Figure 6B). Anti-89RX1 antibodies reacted strongly with 89VP2-p11 and 89VP2-p10, which overlaps with 89VP2-p11 and is part of the RX1 construct, whereas the reactivity with 89VP1-p3 was much lower. No reactivity of anti-89RX1 antibodies to 89VP1-p5 and 89VP1-p9, which are not part of RX1 over the background (i.e., BSA), was observed (Figure 6B). 

Anti-89RX2 antibodies showed strong reactivity to 89VP1-p5, 89VP1-p9 and 89VP2-p11, whereas reactivity to 89VP2-p10 was lower (Figure 6B). No reactivity to 89VP1-p3, which was not part of the construct over the background (i.e., BSA), was found (Figure 6B). 

The anti-89RX1 antiserum neutralized RV89 infection in vitro up to a dilution of 1:64, whereas the neutralizing capacity of the anti-89RX2 antiserum was much lower (i.e., 1:4 dilution) (Figure 6C). In addition, GPαRV89 antibodies neutralized RV infection up to a dilution of 1:6000–1:24,000 (Figure 1). A comparison of the reactivity of the GPαRV89 antibodies with natural RV89 with that of anti-89RX1 and anti-89RX2 antibodies showed a much stronger reactivity (i.e., more than 100-fold) of GPαRV89 antibodies compared to anti-89RX1 and anti-89RX2 antibodies with RV89, but a direct comparison of titers cannot be made due to the different detection of rabbit and guinea pig antibodies in this experiment (Figure 6D). However, the latter experiment would suggest that antibody levels/concentrations rather than antibody specificities and affinities/avidities are responsible for the different neutralization capacity of GPαRV89 antibodies and anti-89RX1 and anti-89RX2 antibodies.

## 3. Discussion

More than 160 RV strains are known, which differ from each other in terms of sequence and binding to different receptors on human cells. RV infections are usually mild and short lasting but can trigger severe exacerbations of asthma and hence can be life-threatening. Accordingly, there is a need for active and passive immunization strategies aiming at either the induction or administration of antibodies which can block the binding of RV strains to their receptors and prevent RV infections. The ability of antibodies to interfere with the RV–receptor interaction can be studied by classical in vitro virus neutralization assays [40], by surrogate assays mimicking the virus–receptor interaction [41] and by in vivo assays based on, for example, mice which are transgenic for the human receptors [42]. Here, we have produced by expression the four RV89 capsid proteins, VP1-VP4, recombinant fragments and synthetic peptides thereof, to map the binding sites of an antiserum which strongly neutralized the infection of human cells by RV89, a representative member of major-group RV-A species. The neutralizing antiserum reacted with an N-terminal and C-terminal fragment of VP1 and the middle portion of VP2, but not with VP3 and VP4, indicating that VP1 and VP2 are important for the binding of RV89 to its receptor ICAM-1. This result is in agreement with our recent finding that antibodies raised against VP1 and VP2, but not against VP3 or VP4, inhibited the binding of RV89 to the ICAM-1 protein in an ELISA-based interaction assay, based the interaction of RV89 with plate-coated recombinant ICAM-1 [41]. Furthermore, the fact that VP4 does not appear on the surface of the three-dimensional structure of the capsid of RV16, which is closely related to RV89, indicates that VP4 is not involved in the RV89–ICAM-1 interaction. However, an earlier study has suggested that antibodies directed to the N-terminal portion of VP4 may be responsible for cross-serotypic virus neutralization [35]. A more detailed mapping of the binding sites of the highly neutralizing antiserum showed that the antiserum reacted with three VP1-derived synthetic peptides and one VP2-derived synthetic peptide, which were part of the independently identified recombinant VP1 and VP2 fragments. In addition, 89VP1-p1 and 89VP1-p3 were located in the N-terminal VP1 fragment, 89VP1-p9 was located within the C-terminal VP1 fragment and 89VP2-p11 was located in the middle VP2-derived recombinant fragment. Furthermore, 89VP1-p1 corresponded to an earlier-defined epitope which is highly reactive with sera from RV-exposed/infected subjects [39] and can be used for species-specific serological diagnosis of a recent RV infection by ELISA or microarray-based chip testing [21,38]. However, 89VP1-p1 does not appear on the surface of the three-dimensional structure of the capsid of RV16 and hence does not seem to be involved in virus neutralization (Figure 5). In fact, we have immunized rabbits with the KLH-coupled 89VP1-p1 peptide but have not obtained neutralizing antibodies (data not shown). Interestingly, VP1-derived peptide 89VP1-p3 contains amino acid residues which have been identified in RV strains (i.e., RV2 and RV14) [31,33] (Table 1), which differ substantially from RV89 (Figure 7). RV2 is a minor-group RV-A strain with a highly different sequence, which even binds to a different receptor (i.e., LDLR) to RV89, whereas RV14 is a major-group RV-B strain which strongly differs in sequence from RV89. Peptide 89VP1-p3 is close to 89VP1-p5, which has been reported to contain amino acids important for virus neutralization (Table 1) [11,29,33]. Both peptides appear in close vicinity on the surface of the structure of the capsid of RV16 (Figure 5), but 89VP1-p5 did not react with the highly neutralizing antiserum. Moreover, the antiserum recognized peptide 89VP1-p9, which is located at the C-terminal fragment of VP1. Strikingly, 89VP1-p9 also identified peptides located at the same positions of the highly different RV2 and R14 strains, which contained amino acids identified to play an important role in virus neutralization (Table 1) [31,33,34]. Finally, the last peptide, 89VP2-p11, also recognized by the RV89-neutralizing antiserum, superimposes with peptides within the VP2 sequences of the highly divergent strains, RV2 and RV14, which have been found to contain amino acids involved in virus neutralization (Table 1) [28,32,33]. Thus, the epitope mapping performed with the RV89-neutralzing antiserum identified corresponding regions with similar locations in the capsid of two highly divergent RV strains, which differ highly in sequence when compared to RV89. In order to obtain information on whether the peptides recognized by the RV89-neutralzing antiserum indeed play a role in virus neutralization, we constructed two recombinant fusion proteins containing the identified peptides fused to HBV-derived preS as a carrier protein. We then immunized rabbits with these two fusion proteins (i.e., 89RX1 and 89RX2) and indeed showed that the rabbit antisera specific for 89RX1 and 89RX2 were able to neutralize the infections of cells by RV89 in vitro. The virus-neutralizing activity of the antisera specific for 89RX1 and 89RX2 was much lower than that of the RV89-neutralizing antiserum, but this seemed to be due to the fact that the antibody levels specific for RV89 in the RV89-neutralizing antiserum were much higher than those of the antisera specific for 89RX1 and 89RX2 (Figure 6). Other reasons for the lower virus-neutralizing capacity of anti-89RX1 and anti-89RX2 antibodies may of course be that that the affinity/avidity of these antisera was lower, because these antisera were obtained by immunization with unfolded proteins obtained by expression in *Escherichia coli* whereas the RV89-specific antiserum was obtained by immunization with the intact RV89 virus. 

Limitations of our study are that it can be difficult to generate broadly neutralizing vaccines because of the differences in sequences of the identified epitope regions among the large number of RV strains and the fact that it may be necessary to include relevant T cell epitopes, which may help to quickly mobilize neutralizing antibody responses. The plans towards the design of an effective vaccine are therefore to create a cocktail of immunogens from representative RV strains, which induces a highly cross-reactive protective antibody response against the identified epitopes that is accompanied by T cell responses. 

Nevertheless, our results point to the possibility that similar areas/ portions of the capsid of even highly divergent RV strains are involved in RV infection and in the binding to different RV receptors ICAM-1 and LDLR, and perhaps even CDHR3. Accordingly, it may eventually be possible to develop broadly neutralizing RV vaccines by incorporating corresponding portions of the identified segments of VP1 and VP2, which are representative for the most commonly occurring RV strains in recombinant immunogens.

## 4. Materials and Methods

### 4.1. Antisera and Detection Antibodies 

Guinea pig RV89-specific antiserum and the corresponding pre-immune serum was purchased from the American Type Culture Collection (ATCC, Manassas, VA, USA). Rabbit antibodies against recombinant 89RX1 and 89RX2 were obtained by immunizing each rabbit three times with 200 µg of the corresponding antigen using Complete Freund’s adjuvant (CFA), followed by Incomplete Freund’s adjuvant (IFA) (Charles River Laboratories, Sulzfeld, Germany). Normal rabbit serum used for control purposes was obtained from a non-immunized rabbit (Genscript, Leiden, The Netherlands). HRP-conjugated goat anti-GP IgG was purchased from Jackson ImmunoResearch Laboratories, Westgrove, PA, USA, and HRP-labelled donkey anti-rabbit antibodies from Amersham, Malborough, MA, USA.

### 4.2. RV89-Derived Recombinant Capsid Proteins, Protein Fragments and Synthetic Peptides

Recombinant RV89 capsid proteins (i.e., 89VP1-89VP4) as well as recombinant maltose binding protein (MBP)-fusion proteins containing 89VP1 and 89VP2 fragments of approximately 100 aa (i.e., 89VP1-PI-PIII; 89VP2-PI-PIII) (Appendix A) were expressed in Escherichia coli and purified by nickel affinity chromatography, as previously described [21,39]. Furthermore, 89VP1- and 89VP2-derived synthetic peptides were produced by solid-phase synthesis with an Fmoc (9-flurorenylmethylmethoxycarbonyl) strategy and 2-(1H-benzotriazol-1-yl)-1,1,3,3-tetramethyluronium hexafluorophosphate (HBTU) activation using two peptide synthesizers (Liberty CEM GmbH, Kamp-Lintfort, Germany, and AAPPTec, Louisville, KY, USA) [21]. After synthesis, peptides were cleaved from the resins using 19 mL trifluoroacetic acid (TFA), 0.5 mL silane and 0.5 mL H_2_O, and precipitated into pre-chilled tertbutylmethylether. Cleaved peptides were purified by reversed-phase high-performance liquid chromatography in a 10–70% acetonitrile gradient using a Jupiter 4 μm Proteo 90 Å, LC column (Phenomenex, Torrance, CA, USA) and an UltiMate 3000 Pump (Dionex, Sunnyvale, CA, USA) to a purity of >90%, as verified by mass spectrometry (Microflex MALDI-TOF, Bruker, Billerica, MA, USA).

### 4.3. SDS-PAGE and Immunoblotting

The reactivity of RV89-specific guinea pig antibodies to the RV89 capsid proteins and fragments thereof was determined by immunoblotting. For this purpose, two micrograms of each antigen were separated by 14% sodium dodecylsulfate polyacrylamide gel electrophoresis (SDS-PAGE) and stained with Coomassie Brilliant Blue (Pierce, Thermo Fischer Scientific, Waltham, MA, USA). Identically prepared gels were blotted onto nitrocellulose membranes (GE Healthcare Life Sciences, Amersham, UK). Blotted membranes were then incubated with RV89-specific guinea pig antisera diluted to 1:1000 and bound IgG antibodies were detected with 1:5000 diluted HRP-conjugated goat anti-GP IgG (Jackson ImmunoResearch Laboratories, PA, USA). The binding of the antibodies to the blotted antigens was visualized using the Amersham ECL Prime Detection Reagent (GE Healthcare Life Sciences, Amersham, UK). Unrelated proteins such as the maltose binding protein (MBP) and a PreS fusion protein (i.e., PreS-P1-P5) [43] were used for negative control purposes.

### 4.4. Enzyme-Linked Immunosorbent Assay (ELISA)

Recombinant capsid proteins, protein fragments and synthetic peptides of RV89 as well as purified RV89 were coated (2 µg/ml in 100 mM bicarbonate/carbonate coating buffer) onto 96-well ELISA plates (Maxisorp, Nunc, Thermo Fisher Scientific, Waltham, MA, USA). After overnight (ON) incubation at 4 °C, plates were washed with phosphate-buffered saline (PBS) containing 0.05% Tween-20 (PBS-T) and blocked with 2% bovine serum albumin (BSA)/PBS-T for 2 hours at room temperature. Different dilutions of guinea pig (GP) or rabbit antisera in 0.5% PBS/T were added to the plates and incubated at 4 °C overnight. After washing with PBS-T, bound GP or rabbit antibodies were detected with either 1:5000 diluted (in 0.5% BSA/PBS-T) HRP-conjugated goat anti-GP IgG (Jackson ImmunoResearch Laboratories, West Grove, PA, USA) or 1:2000 diluted HRP-labelled donkey anti-rabbit antibodies (Amersham, Malborough, MA, USA), respectively. Optical density values (i.e., O.D.) corresponding to bound IgG were recorded in an ELISA reader (Tecan Group Ltd., Männedorf, Switzerland).

### 4.5. In Vitro Virus Neutralization Assays

Virus neutralization assays were performed as described [37]. Ohio-strain HeLa cells (1.3 × 10^4^ per well) were seeded into 96-well plates (Costar, Corning, NY, USA) using Minimal Essential Medium (MEM) containing 2% FCS, 30 mM MgCl_2_ (i.e., infectious medium). HeLa cells were then grown at 37 °C with 5% CO_2_ to a confluence of approximately 80–90%. Then, 50-fold TCID_50_ of RV89 in Infection Medium was mixed with serially diluted RV89-specific guinea pig (i.e., GPαRV89), normal guinea pig serum or rabbit (i.e., anti-89RX1 and anti-89RX2) antisera or rabbit pre-immune sera, and incubated for at least 3 h at 37 °C. HeLa cells were then overlaid with these mixtures and incubation was continued at 34 °C for 3 days. For control purposes, cells were incubated with either medium or RV89 alone. Afterwards, the medium was removed and cells were stained with 0.1% crystal violet (Sigma-Aldrich, St. Louis, MO, USA) for 10 min. After washing with water, the plates were dried and the stained cells were dissolved in 30 μL 1% SDS under shaking for 1 h. Cell protection was quantified at A560 in a TECAN Infinite F50 ELISA reader with the integrated software i-control 2.0 (Tecan Group Ltd., Männedorf, Switzerland). 

### 4.6. Construction of Recombinant PreS-Carrier-Based Fusion Proteins, 89RX1 and 89RX2

Both 89RX1 and 89RX2 are recombinant fusion proteins in which hepatitis B-derived preS serves as a carrier molecule for attached peptides from unrelated proteins. It has been developed for the construction of vaccines for allergen-specific immunotherapy with the goal of enhancing immune responses to the peptides through T cell help derived from the carrier molecule [43,44]. DNA regions coding for the identified RV89-derived epitopes were fused to the N- and/or C-termini of the PreS region of HBV subtype adw2 (GenBank: AAT28735.1) as previously described, and as shown in Figure 6 [43,44]. Genes encoding the fusion proteins with a C-terminal hexahistidine tag were codon-optimized for expression in *E. coli*, synthesized (GenScript, Piscataway, NJ, USA) and inserted into the NdeI/XhoI sites of plasmid pET-27b (Novagen, Darmstadt, Germany). The DNA sequences were checked by means of restriction enzyme analysis of the midi-prep plasmid DNA (Promega, Madison, Wish) with NdeI/XhoI (Roche, Mannheim, Germany) and by automated sequencing of both DNA strands (Eurofins Genomics, Ebersberg, Germany).

### 4.7. Expression, Purification and Characterization of Recombinant PreS-Carrier-Based Fusion Proteins 

Recombinant 89RX1 and 89RX2 were expressed in *E. coli* strain BL21(DE3) (Stratagene, La Jolla, San Diego, CA, USA). After the induction of protein expression by adding 1 mM isopropyl-β-D-thiogalactopyranoside, cells were cultured for approximately 3–4 h at 37 °C and then harvested by centrifugation. His-tagged preS fusion proteins were then purified by means of nickel affinity chromatography and dialyzed against 10 mM NaH_2_PO_4_ as described [44]. The biochemical properties of the recombinant preS fusion proteins were calculated from their amino acid sequence using ProtParam (http://web.expasy.org/protparam/ (accessed on 1 April 2022)). Purified recombinant proteins were analyzed by SDS-PAGE under reducing as well as non-reducing conditions. Gels were stained with Coomassie Brilliant Blue. Molecular masses of recombinant proteins were also confirmed by mass spectrometry (Microflex MALDI-TOF, Bruker), as described [44]. Circular dichroism (CD) spectra were measured at a protein concentration of 0.1 mg/mL on a Jasco J-810 spectropolarimeter (Japan Spectroscopic Co., Tokyo, Japan). Spectra were recorded at 25 °C between 190 and 260 nm, with a resolution of 0.5 nm, and at a scan speed of 50 nm/min. The final spectra were baseline-corrected by subtracting the corresponding buffer spectra obtained under identical conditions. The results were expressed as mean residue ellipticity (Ø) at a given wavelength [45].

## Figures and Tables

**Figure 1 ijms-23-05113-f001:**
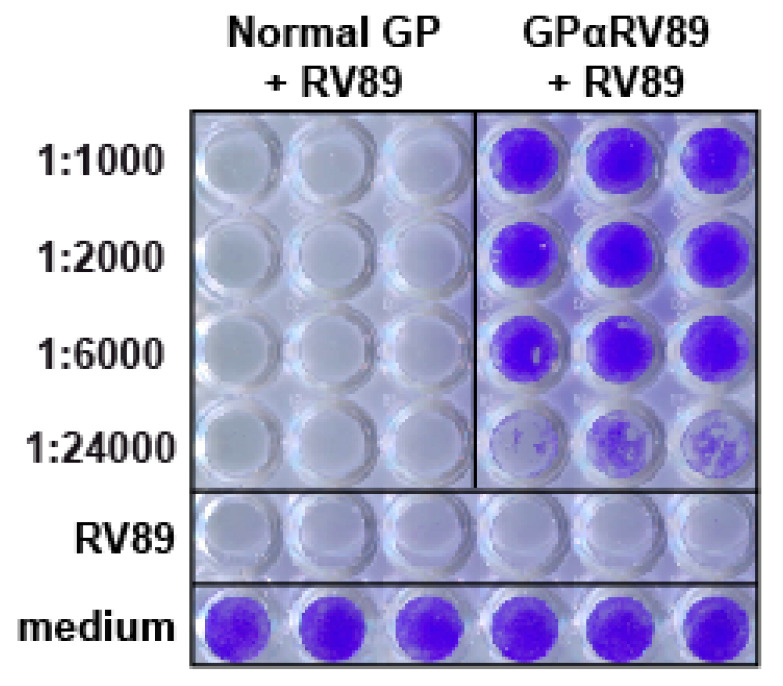
Neutralization of RV infection of HeLa cells with an anti-RV89 guinea pig antiserum. Shown are HeLa cells infected with 50-fold TCID_50_ of RV89 which had been preincubated with different dilutions of normal guinea pig serum or of the anti-RV89 antiserum, respectively. For control purposes, cells were only incubated with medium or medium containing 50-fold TCID_50_ of RV89. Cells were stained with 0.1% crystal violet. Stained wells contain live/protected cells, while in white-unstained wells, cells are infected/dead.

**Figure 2 ijms-23-05113-f002:**
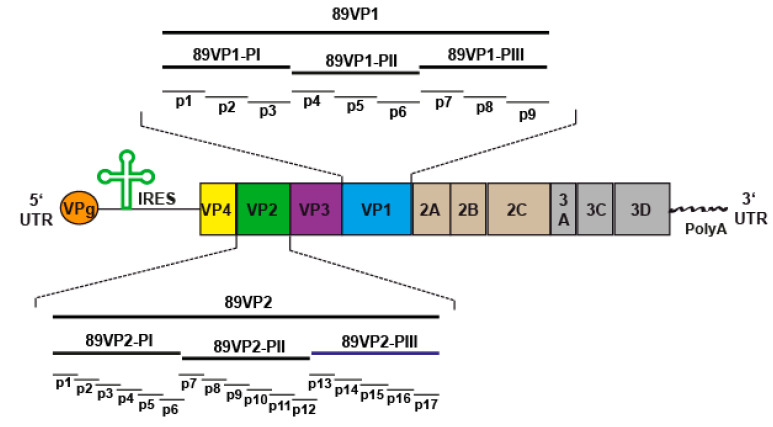
Schematic representation of the rhinovirus genome encoding a polyprotein. All structural (VP1, VP2, VP3, VP4) and non-structural proteins (2A, 2B, 2C, 3A, 3C and 3D) within the polyprotein are indicated. The 5′ UTR containing the IRES and the VPg sequences as well as 3′ UTR containing a poly(A) tail are also shown. The magnifications show the three recombinant fragments of VP1 (top) and VP2 (bottom) (i.e., PI, PII, PIII) as well as VP-derived synthetic peptides (i.e., VP1: p1–p9; VP2: p1–p17). IRES—internal ribosome entry site; RV—rhinovirus; UTRs—untranslated regions; VPg—viral protein genome-linked.

**Figure 3 ijms-23-05113-f003:**
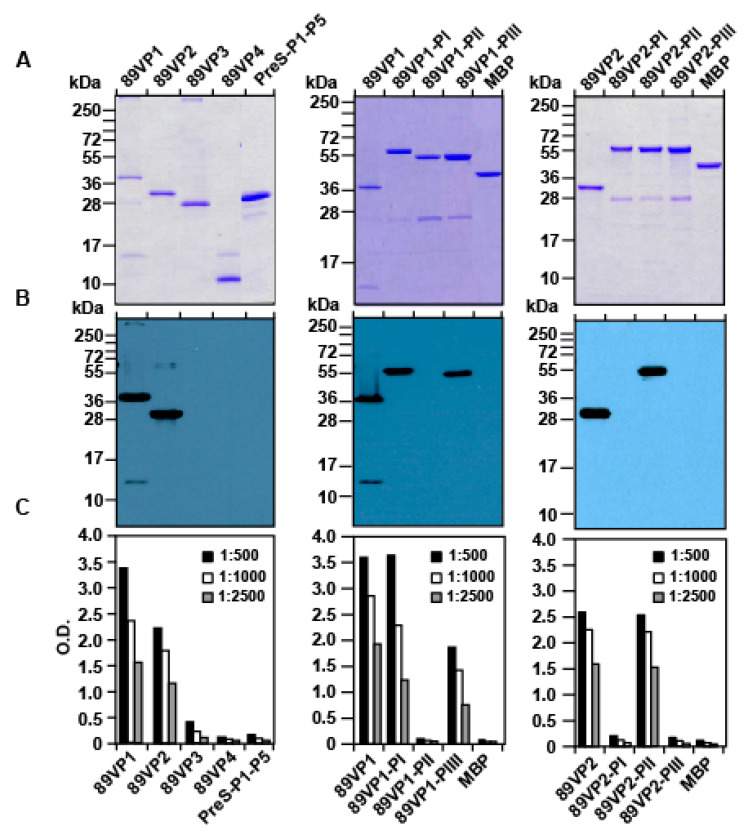
Reactivity of anti-RV89 guinea pig IgG antibodies to recombinant RV89-derived capsid proteins and fragments thereof. (**A**) Coomassie blue-stained SDS-PAGE containing purified viral capsid proteins: 89VP1-89VP4 (**left** panel); VP1 and VP1 fragments: 89VP1-PI-PIII (**middle** panel); and VP2 and VP2 fragments: 89VP2-PI-PIII (**right** panel). PreS-P1-P5 and MBP represent negative control proteins. Molecular weights in kDa are indicated on the left margins. (**B**) IgG reactivity of the anti-RV89 guinea pig antibodies to nitrocellulose-blotted proteins from A. (**C**) IgG reactivity of three different dilutions (1:500, 1:1000, and 1:2500) of anti-RV89 guinea pig antibodies to RV antigens (*x*-axes) as measured by ELISA. Bound IgG antibodies are expressed as optical density values (OD, *y*-axes).

**Figure 4 ijms-23-05113-f004:**
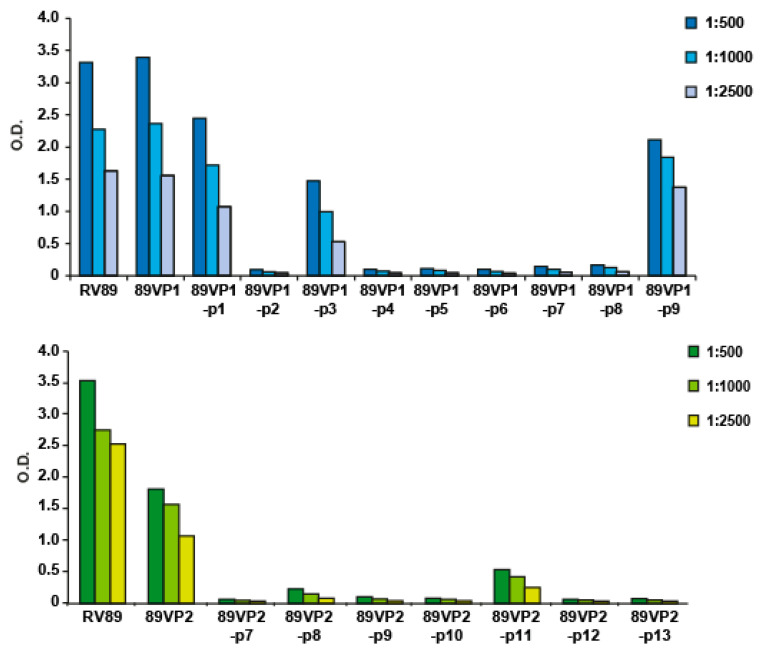
Reactivity of three dilutions of anti-RV89 guinea pig IgG antibodies to purified RV89, recombinant RV89 capsid proteins and synthetic peptides (*x*-axes). Bound IgG antibodies correspond to optical density values (O.D., *y*-axes). PreS-P1–P5 as well as MBP were used as negative controls.

**Figure 5 ijms-23-05113-f005:**
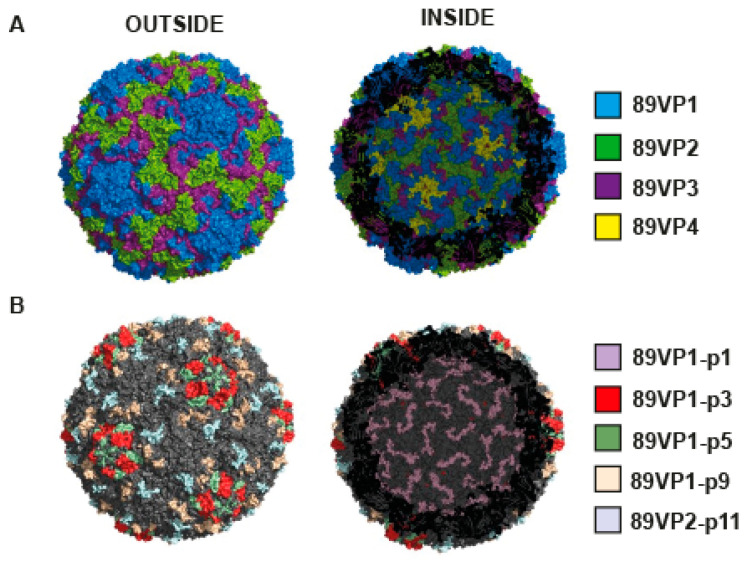
Localization of VP1- and VP2-derived peptides in the model of the three-dimensional structure of the capsid of RV16. Outer (**left** images) and inner (**right** images) surface representations with (**A**) VP1–4 and (**B**) with VP1- and VP2-derived peptides highlighted in different colors.

**Figure 6 ijms-23-05113-f006:**
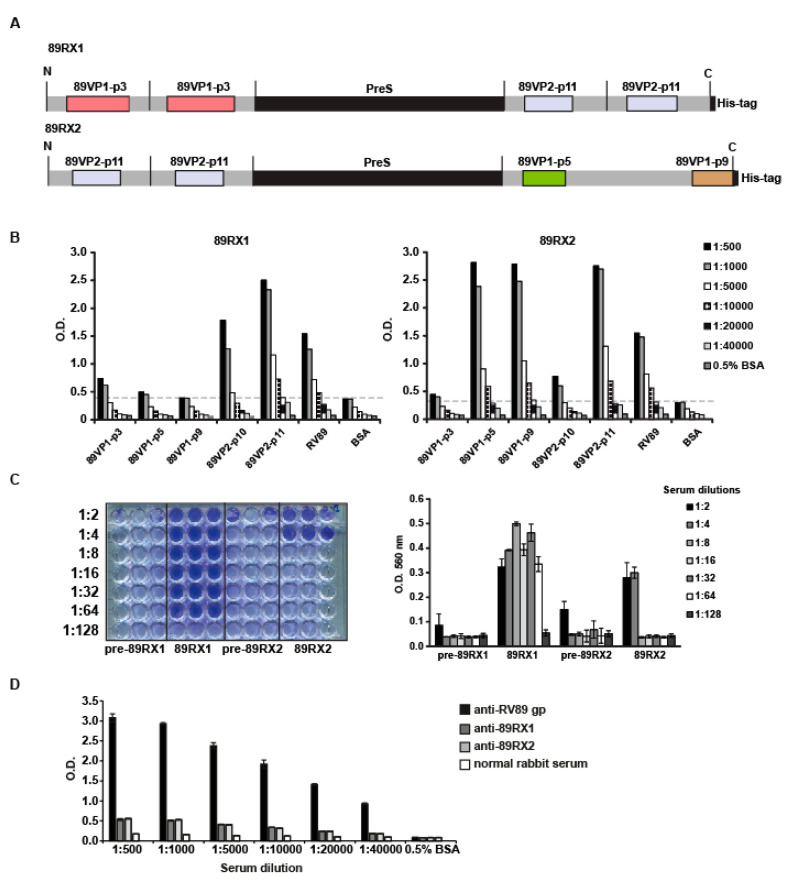
Recombinant PreS-fusion proteins (89RX1, 89RX2) containing VP1- and VP2-derived peptides. (**A**) Scheme of 89RX1 (**upper** panel) and 89RX2 (**lower** panel). The PreS carrier protein is colored in black and VP1- and VP2-derived peptides are indicated in the same colors as in Figure 5. (**B**) Reactivity of rabbit anti-RX1 and anti-RX2 IgG antibodies (*y*-axes: OD values correspond to bound antibodies) to RV89- and RV89-derived antigens and peptides (*x*-axes). (**C**) Inhibition of RV89 infection of HeLa cells by serially diluted (1:2 to 1:128) anti-89RX1, anti-89RX2-specific antisera and the corresponding pre-immune sera. The RV89 and medium alone were used as controls. Shown are cells stained with 0.1% crystal violet for viability (**left** panel) or optical density values (O.D., *y*-axis), measured after the dissolving of the viability dye (**right** panel). (**D**) The RV89-specific IgG reactivity of different dilutions of the anti-RV89 guinea pig antiserum or of anti-89RX1 and of anti-89RX2 rabbit antisera measured by ELISA. Optical density (OD) values (*y*-axis) correspond to bound RV89-specific IgG.

**Figure 7 ijms-23-05113-f007:**
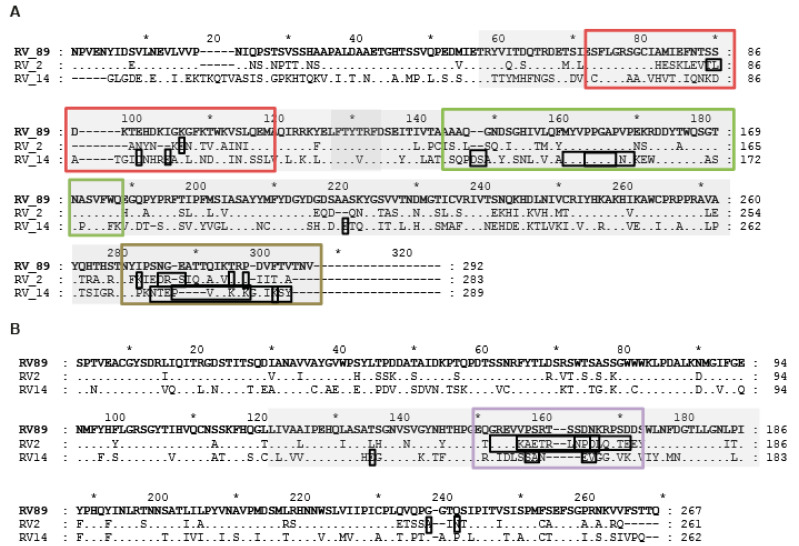
Sequence comparison of RV89 with two RV strains (RV2, RV14) for which epitopes involved in virus neutralization have been determined. Shown are amino acid sequence alignments of (**A**) VP1 and (**B**) VP2 capsid proteins of RV89 (major receptor group, RV-A) on top with RV2 (minor receptor group, RV-A) and RV14 (major receptor group, RV-B). The RV89-derived peptides identified as potential epitopes involved in virus neutralization in our study are boxed (red: 89VP1-p3; green: 89VP1-p5; brown: 89VP1-p9; purple: 89VP2-p11) and peptide-containing fragments included in 89RX1 and 89RX2 are highlighted in grey. Amino acids boxed in black were previously reported to be involved in virus neutralization (Table 1). Left margin indicates strain number; dots and dashes represent identical amino acids and gaps, respectively.

**Table 1 ijms-23-05113-t001:** Previously reported capsid proteins or epitopes thereof involved in virus neutralization for minor- and major-group RV strains.

RV Strain	Identified Epitope/Antigen	AA Residue/s (AA Position)	Methodology	Ref.
**RV2** Group A minor	**VP1**	TL (85-6) E (92) K (264) DRS (267-9) T (276) P (278)	RV2 escape mutants (n = 51) were identified using 14 neutralizing monoclonal antibodies. Amino acid substitutions among the 51 mutants were identified.	[31]
	**VP2**	NP (163-4) A (236) N (238)		
	**VP2**	GREVKAETRLNPD (153-64)	Deletions at the 3’-end gene of VP2 using Bal-31 nuclease were created. The polypeptides were probed using a neutralizing monoclonal antibody (mAb 8F5).	[32]
	**VP2/NIm-II**	VKAETRLNPDLQPTE (156-70)	Rabbit anti-peptide antibodies (at 1:4 dilution) were used to determine neutralizing epitopes.	[28]
**RV-14** Group B major	**VP1**	MYVPPGAPNP (151-60)	Amino acid sequences were postulated as potential receptor-binding sites.	[11]
	**VP1**	PPGA (154-7)	A rabbit polyclonal anti-peptide antiserum was used to neutralize RV14.	[29]
	**VP1/NIm-IA**	D (91) E (95)	Approach was the same as Appleyard et al. (1990) (30). Thirty-five RV14-neutralizing monoclonal antibodies were used to identify 62 escape mutants.	[33]
	**VP1/NIm-IB**	Q (83) K (85) DS (138-9)		
	**VP1/NIM-II**	E (210)		
	**VP1/NIm-III**	K (287)		
	**VP2/NIm-II**	SA (158-9) EV (161-2) E (136)		
	**VP1/NIm-IV**	PVIKKRK (275-85)	Residues were discovered via a molecular evolution experiment/VP1 gene shuffling.	[34]
	**VP1/NIm-IV**	NTEPVIKKRKGDIKSY (272-91)	Residues were discovered via a molecular evolution experiment/VP1 gene shuffling.	[34]
	**VP4**	GAQVSTQKSGSHENQNILTNGSNQTFTVINY (1-31)	Antibodies generated to N-terminal VP4 fragment of RV14 showed cross-serotypic neutralization.	[35]
**RV16** Group A major	**VP0**	Consensus sequence for VP4 and VP2	Recombinant VP0 (VP4 + VP2) was used for the immunization of mice. Antibodies induced by VP0 immunogen enhanced neutralizing antibody responses to heterologous virus infection.	[36]
**RV89** Group Amajor	**VP1**	GenBank: AY355270	Recombinant VP1 proteins of RV89 and RV14 were used to induce polyclonal antibodies, which exhibited cross-neutralization.	[37]

## Data Availability

Data can be obtained upon request from the corresponding author.

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
