# Peer review of "Identification of Epitopes on Rhinovirus 89 Capsid Proteins Capable of Inducing Neutralizing Antibodies"

_ijms, 2022, doi:10.3390/ijms23095113_

Round 1

Reviewer 1 Report

In this study the authors set out to identify RV89 epitopes that associate with neutralising antibodies. The paper has nice flow and good experimental approach. The findings are interesting but the advances to RV vaccine research are limited. I have a few minor comments:

  1. Although briefly mentioned, could the authors elaborate on the selection of RV strain, used in the study? Have you used any other RV strain in the assays?
  2. Have you attempted to use the anti-RV89 antisera for other closely related, as well as divergent strains and assess virus neutralisation?
  3. I acknowledge that the aim of the study was the identification of epitopes that induce neutralising antibodies, but this is only a fraction of protective immunity against viruses. Other epitopes- maybe located inside the capsule- will be recognised by T cells. Such epitopes (conserved or not across strains) could contribute to protective immunity against infection through the function of anti-viral specific T cells. The role of such cells should not be neglected, as also could be induced by effective vaccines, and therefore worth being mentioned in the discussion.
  4. Could the authors mention the limitation of the study and also what are the future plans of this work?
  5. RV is a mild infection and this reduces the interest for vaccine development. it would be interesting to put RV infection in the context of respiratory bacterial co-infections, and mention the impact of it on resp disease severity/outcome.

Reviewer 2 Report

This manuscript by Valenta et. al. developed epitopes recognized by a highly neutralizing antiserum in the capsid of RV89 strain which belongs to the major group A virus species. Identified peptide epitopes were then expressed as fusion proteins together with hepatitis virus B (HBV)-derived preS to raise antibodies capable of neutralizing in vitro the infection of cultured human cells by RV89. The whole manuscript was well-organized, and the information provided in this study and the experimental methodology are interesting. Hence, I recommend its publication in IJMS after a minor revision with the following comments addressed.

  1. Improve the resolution of figure 1
  2. The conclusion looks fine, and the main limitation also should be discussed as well.
  3. There are some grammatical errors in this manuscript such as continuously forgetting to add ‘a’ or ‘the’ before a specific word which limits the clarity of the author’s writing. Check the language issues.
